# Mapping factors influencing initiation of antiretroviral treatment among adolescents living with HIV/AIDS in sub-Saharan Africa: A scoping review protocol

**Palesa Grace Likoti**[¤]*, **Desmond Kuupiel**, **Nelisiwe Khuzwayo**

College of Health Sciences, University of KwaZulu-Natal, Durban, South Africa

☯ These authors contributed equally to this work.
¤ Current address: College of Health Sciences, School of Nursing and Public Health, University of KwaZulu-Natal, Durban, South Africa
* parisgracelikoti@gmail.com

**Data Availability Statement:** No datasets were generated or analysed during the current study. All

## Abstract

### Background

Since the Start Free, Stay Free, and AIDS-Free launch, UNAIDS targets intended to promote interventions to prevent HIV transmission and promote access to ART among adolescents and children, of which none were achieved in 2020. In the sub-Saharan African region, the number of adolescents initiated on ART drugs remained consistently low, with approximately100 000 adolescents succumbing to AIDS-related causes in 2022. Although HIV prevalence among adolescents had been reduced, several HIV- positive adolescents died without being initiated on ART drugs. Therefore, this scoping review protocol aims to map factors influencing the initiation of ART drugs among adolescents living with HIV in sub-Saharan Africa.

### Methods

The methodological framework for scoping reviews will guide this scoping protocol. A search strategy will be used to search literature in electronic databases, including EBSCOhost (PubMed/MEDLINE), Google Scholar, Science Direct, Scopus, BioMed Central, and the World Health Organization library for citations and literature using keywords and the Medical Subjects Heading (MeSH). The electronic databases will be supplemented by hand-searching references on the included studies. The search will be from Jan 01, 2012, to Dec 31, 2022. Articles will be searched and assessed for eligibility by two screeners uploaded on the Endnote software, and duplicates will be identified and removed before the abstract screening. The two screeners will assess the eligibility of the abstracts and the complete articles of the selected studies using the inclusion and exclusion criteria. A third screener will intervene when there is a lack of consensus between the two screeners. The selection process will be documented by following and using the PRISMA flow diagram (Fig 1). A thematic content analysis will present a narrative account of the extracted data.

relevant data from this study will be made available upon study completion.

**Funding:** PG Likoti was funded by the University of KwaZulu-Natal, College of Health Sciences PhD scholarship funding. The funders had no role in the analysis or writing of this paper. https://snph.ukzn.ac.za/.

**Competing interests:** The authors have declared that no competing of interest exits.

**Abbreviations:** ART, Antiretroviral Therapy; AIDS, acquired immunodeficiency syndrome; HIV, Human Immunodeficiency Virus; UNAIDS, The Joint United Nations Programme on HIV/AIDS; PRISMA, Preferred Reporting Items for Systematic Reviews and Meta-Analyses; SSA, sub-Saharan Africa.

## Discussion

The results of this review will identify and describe factors influencing the initiation of Antiretroviral treatment among adolescents living with HIV in the Sub-Saharan African region. The findings will guide future research and inform tailored interventions and strategies for initiating ART among adolescents.

## Trial registration

Open Science Framework. https://doi.org/10.17605/OSF.IO/RNF2T.

## Background

Worldwide, an estimated 1.8 million adolescents between the ages of 10 and 19 were reported to be living with HIV by the end of 2020 [1]. Approximately 90% of them reside in the sub-Saharan Africa, which is the epicentre of HIV [2]. Globally, approximately 54% of adolescents aged 10–19 received ART in 2021 [1]. There was no significant difference in ART coverage between adolescent girls (58%) and boys (60%), respectively [3].

Among them, approximately 110,000 children and adolescents died from AIDS-related complications worldwide in 2021 [4]. Since there is no cure for HIV/AIDS, primary interventions significantly prevent HIV infections. Although these interventions are prevalent in sub-Saharan Africa, HIV infection among adolescents aged 10–19 is still high, with an estimated 480,000 adolescents newly infected with HIV in 2022 alone [5]. HIV treatment has been scientifically proven to save lives if accessed early and used appropriately, yet several adolescents succumb to AIDS-related complications annually [6].

In sub-Saharan Africa, numerous interventions promote access to HIV treatment, including primary prevention and HIV treatment literacy; most of these interventions target older people [7]. A study conducted in South Africa revealed sizable gaps in the uptake of critical HIV services targeting young people [8]. These are congruent with a study conducted in Zimbabwe that found a growing number of new infections among adolescents and young people [9]. Hence significantly shaping the HIV epidemic. Nevertheless, most HIV programs and other health services have been primarily orientated toward adults, with limited commitment to the specific needs of children and adolescents [10].

While programmes to prevent new HIV infections among people with HIV have been hugely successful in recent years, low HIV testing, diagnosis and treatment initiation has been slow and more difficult to achieve among young people (10–19 years) in sub-Saharan Africa [11]. To close the gaps in HIV treatment and prevention, the Joint United Nations Programme on HIV/AIDS (UNAIDS) adopted ambitious, fast-track targets aimed at getting 90% of people living with HIV to know their HIV status, 90% of people to receive treatment and 90% of people on treatment to be virally suppressed. Driven by the concern that these targets were not achieved in 2020, the targets have been revised and increased to 95% by 2025. Many countries in sub-Saharan Africa are far from achieving 95-95-95 by 2025 [12].

Treatment strategies of 95-95-95 targets revealed various challenges prohibiting meaningful access to treatment and favourable health outcomes among young people [13]. Some challenges include unprepared health systems that address unique adolescents' needs and factors such as stigma and discrimination, resulting in adolescents' reluctance to take ART [14].

Therefore, poor ART coverage among adolescents is limited to access to HIV testing and non-adherence, which decreases viral load suppression among adolescents compared to adults [15].

A framework was launched in 2015 by the Joint United AIDS, aiming to end HIV/AIDS among children, adolescents, and young women, specifically focusing on sub-Saharan Africa [16]. The target set to be reached by 2020 was to Start Free, Stay Free and be AIDS-Free; though none was met in 2020. For instance, the stay-free target of reducing the number of new HIV infections among adolescents and young women (aged 10–19 years) to less than 100,000 fell short. The World Health Organisation reported approximately 1 100 new HIV infections daily among adolescents aged 10–19 in 2022 in sub-Saharan Africa [17]. While estimates of new HIV infections are staggering, the AIDS-free target aiming at providing 1.4 million children aged 0–14 years and one million adolescents(aged-10-19) HIV treatment was not reached, and more than half (740 00) children were reached by 2020, only 480 000 adolescents were initiated in ART [18].

Since these targets were not reached, countries implemented HIV interventions supporting Start Free, Stay Free, AIDS-Free targets set to be reached by 2030 [19]. Hence, this study aims to map evidence of factors influencing ART initiation among adolescents aged 10–19 years living with HIV in sub-Saharan Africa. The study's findings recommend that healthcare services revise ART initiation guidelines that will prioritise adolescents' needs and improve healthcare service delivery to accommodate adolescents.

## Methods

This scoping review aims to synthesise evidence on factors influencing ART initiation among adolescents living with HIV/AIDS in sub-Saharan Africa. This scoping review forms part of a PhD study approved by the Biomedical Research Ethics Committee (BREC) of the University of KwaZulu-Natal (BREC Ref No: 0001784/2020). Patients and the public were not involved in the design of this study. This scoping review will adopt a framework that was proposed by Arksey and O'Malley, which involves following the steps: Identification of research questions, identification of relevant studies, Selection of the studies, extracting and charting the data, Collating, summarising, and reporting the data [20]. The steps are described in detail below:

### Step1. Identifying research question

The main proposed research question for this scoping review is "What evidence on the factors influencing ART initiation exists among adolescents living with HIV in sub-Saharan Africa?"

The sub-questions are:

1. What evidence on the individual factors influencing adolescents' initiation of ARTs in sub-Saharan Africa exists?

2. What evidence exists on family and community factors influencing adolescent initiation of ARTs in sub-Saharan Africa?

3. What evidence of health system factors influencing adolescent ART initiation in sub-Saharan Africa exists?

### Stage 2: Identification of the relevant studies

We will search for relevant literature focused on adolescents and HIV treatment from PubMed, Web of Science, EBSCOhost (Academic search complete, CINAHL and MEDLINE) S1 Table, Google Scholar, and the World Health Organization library database, as well as other relevant websites that have literature on this study. We will also manually assess reference lists

for relevant articles. The team will review and assess the results to ensure the validity of the search strategy. Further, all the results from databases and manual searches will be exported to the Endnote X 9 library, and duplicates will be identified and removed. The search was conducted on Dec 31, 2022. The Population-Concept-Context (PCC) framework S2 Table will be used to determine the eligibility of the research question.

## Eligibility criteria

Studies to be included in the review must meet the following criteria:

- Studies with study participants aged 10–19 years.

- Studies with evidence of ART treatment among adolescents.

- Studies with evidence on ART initiation interventions among adolescents.

- Studies with evidence about HIV-positive adolescents.

- Studies published from Jan 01, 2012, to Dec 01 2022.

- Peer-reviewed literature, grey literature, government documents, policy briefs, systematic reviews, and meta-analyses.

- Studies conducted in sub-Saharan African countries

- Studies on original research articles reporting on the factors contributing to ART initiation among adolescents living with HIV.

- Studies published in peer-reviewed journals and conducted in sub-Saharan Africa, using any study design that addresses ART initiation among adolescents living with HIV.

## Exclusion criteria

The following exclusion criteria will include:

- Studies involving HIV-negative adolescents and young people

- Studies focused on HIV screening and/or testing

- Studies focused on HIV viral suppression.

- Articles published in other languages other than English

## 3. Selection of eligible studies

The selection of the studies will be summarised using a PRISMA flow chart Fig 1. The researchers will pilot the search strategy to assess its appropriateness on the selected electronic databases and the keywords of our study. The principal investigator will conduct the title screening, and two reviewers will do the abstract screening independently. The full-text article screening will be done independently, based on the eligibility criteria. When a disagreement arises between two reviewers, the third reviewer will decide. Where there are no complete copies of articles, the researchers will seek assistance from the University of KwaZulu-Natal (UKZN) library services and contact authors to request complete copies of articles attainable within the UKZN library database. The articles will be assessed to ensure we can answer the research question. Empirical studies with mixed methods, qualitative and quantitative studies will be included. The other types of literature included are review articles, commentary and dissertations (Fig 1).

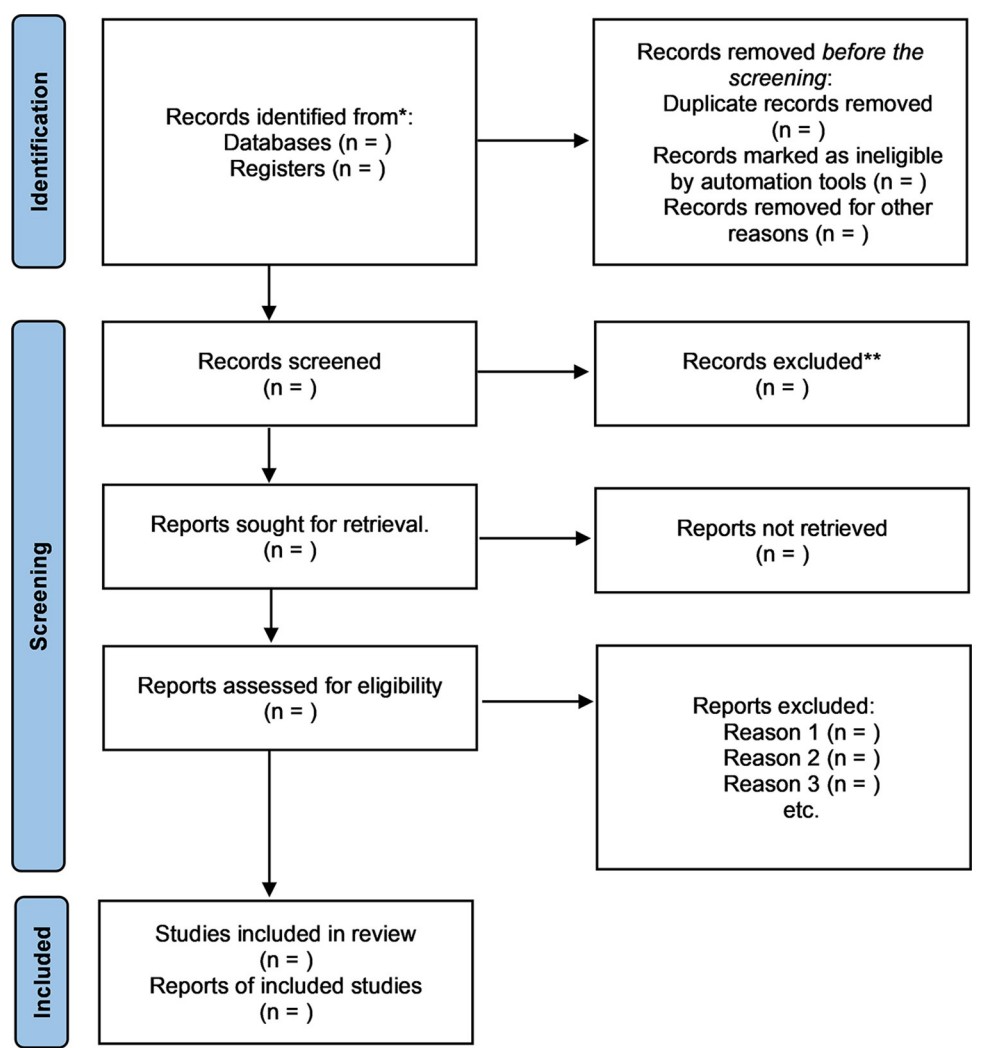

**Fig 1. This is the PRISMA chart diagram.**

**Quality assurance of the study.** To assess the quality assurance of the study, we will use the Mixed Method Appraisal tool version 2020 (MMAT) S3 Table [21]. This tool will be employed to examine the quality of an article by looking at the following aspects: the appropriateness of the aim of the study, if it is suitable, the methodology used, the study design, the recruitment of participants, the data collection process, the data analysis, presenting of findings, the discussions by the authors and conclusions. Additionally, the authors will cross-examine the themes obtained and critically examine their relationship in line with the study's research questions. Furthermore, this will be done by examining the meanings of the findings and their relation to the aim of the study and the implications for future studies.

## Stage 4: Charting the data

Data from included studies will be captured electronically using a data charting form. The extracted data will include all the variables that can assist the team in answering the research questions, including the author with date, study design, population, interventions identified, study setting, the significance of findings, and conclusions for the interventions' primary and secondary outcomes S4 Table.

### 5: Collating, summarising, and reporting the result

We will use NVIVO software version 10 to code the data from the included studies following the below process:

- Coding data from the selected articles.

- categorising the codes into themes.

- displaying the data.

- identifying critical patterns in the data and identifying sub-themes.

- Summarising and synthesising.

A thematic content analysis will present a narrative account of the extracted data around the following themes: individual factors, family and community factors and health system factors influencing the initiation of antiretrovirals among adolescents. The results will be described about the research question and the study's overall purpose.

## Discussion

This scoping review aims to identify and describe factors influencing the initiation of HIV treatment among adolescents. The evidence from the current review may provide a framework that can improve initiating adolescents to ART regime and develop health promotion strategies to enable adolescents living with HIV to seek ART drugs early.

Despite increasing evidence that showed the importance of initiating ART early, high mortality and poor retention prevail among HIV-infected adolescents in sub-Saharan Africa, and young people are still confronted with specific challenges at all stages of the HIV care pathways of diagnosis and ART initiation [22].

The United Joint Nations report on HIV estimates revealed that the highest number of adolescents living with HIV globally are in sub-Saharan Africa [23]. Given the existing behavioural interventions and HIV awareness campaigns promoting strategies, young people can use them to protect themselves from HIV infection [24].

It is crucial to consider behavioural and environmental factors in this region that increase the HIV risk for adolescents exposed to sexual assault and some incidences resulting in HIV transmission to monitor the HIV response among this population [25].

While there is commendable coverage of HIV prevention of mother-to-child transmission among pregnant women living with HIV, some children are born with HIV [26]. Regrettably, understanding the risks and vulnerabilities of children transitioning into adolescence in this region is limited. Less is known about adolescents and how to address their needs for ART initiation and treatment services compared to infants and adults.

Interventions aiming at promoting and enrolling adolescents in ARTs are crucial in saving lives. Most countries in sub-Saharan Africa are implementing a start-free, stay-free, AIDS-free framework to fast-track the eradication of HIV/AIDS among adolescents [19]. Therefore, this study's findings may benefit stakeholders involved in implementing ART interventions and providing other health services targeting adolescents.

## Definition of terms

**Adolescent.** The World Health Organization has defined adolescence as a unique development phase between childhood and adulthood from 10–19.

**Antiretroviral Therapy (ART).**   This is a collection of three or more drugs given to people infected with HIV to suppress the viral load in their blood. These three drugs suppress HIV to prevent it from replicating.

**Adherence.**   The extent to which a patient's behaviour coincides with the prescribed health care regimen agreed upon through a shared decision-making process between the patient and the health care provider.

**Intervention.**   This is defined as an act or strategy developed for, with, or on behalf of individuals or populations to determine, improve, maintain, promote or modify health systems, functioning, or conditions.

**ART initiation.**   This is often defined as a non-emergency intervention and various approaches used to prepare people to begin treatment.

**Influencing.**   Is the capacity to affect character, behaviour, or development to achieve a particular goal.

**Healthcare.**   Is defined as taking preventative or necessary medical procedures to improve an individual's well-being.

## Supporting information

**S1 Checklist. PRISMA-P (Preferred Reporting Items for Systematic review and Meta-Analysis Protocols) 2015 checklist: Recommended items to address in a systematic review protocol\*.**
(DOCX)

**S1 Table. PUB MED search strategy.**
(TIF)

**S2 Table. Population-concept-context.**
(TIF)

**S3 Table. Mixed method appraisal tool version 2020.**
(TIF)

**S4 Table. Data charting form.**
(TIF)

## Acknowledgments

The authors would like to acknowledge the University of KwaZulu-Natal.

## Author Contributions

**Conceptualization:** Palesa Grace Likoti, Nelisiwe Khuzwayo.

**Data curation:** Palesa Grace Likoti.

**Formal analysis:** Palesa Grace Likoti.

**Funding acquisition:** Palesa Grace Likoti.

**Investigation:** Palesa Grace Likoti.

**Methodology:** Palesa Grace Likoti, Nelisiwe Khuzwayo.

**Project administration:** Palesa Grace Likoti, Nelisiwe Khuzwayo.

**Resources:** Palesa Grace Likoti.

**Software:** Palesa Grace Likoti.

**Supervision:** Desmond Kuupiel, Nelisiwe Khuzwayo.

**Validation:** Palesa Grace Likoti, Desmond Kuupiel.

**Visualization:** Palesa Grace Likoti, Nelisiwe Khuzwayo.

**Writing – original draft:** Palesa Grace Likoti.

**Writing – review & editing:** Palesa Grace Likoti, Desmond Kuupiel, Nelisiwe Khuzwayo.

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
