## [Decision Letter · Decision Letter 0]

12 Sep 2023

PONE-D-23-19003Mapping factors influencing initiation of antiretroviral treatment among adolescents living with HIV/AIDS in sub-Saharan Africa: A scoping review protocol.PLOS ONE

Dear Dr. Likoti,

Thank you for submitting your manuscript to PLOS ONE. After careful consideration, we feel that it has merit but does not fully meet PLOS ONE’s publication criteria as it currently stands. Therefore, we invite you to submit a revised version of the manuscript that addresses the points raised during the review process.

The manuscript has improved; however, there still are weaknesses that need to be addressed before publication. First, the manuscript should be consistent to use sub-Saharan Africa in the text instead of using interchangeably sub-Saharan region and SSA. Second, and this is crucial, the "Limitations of the study" are weakly written. It should answer the following question: "How this paper stands out in the area?". As it stands now, it is unclear of its relevance in the field. Third, it is also unclear regarding the age group. Authors refer to young people as 10-24 years or 15-24 years. This needs clarification and consistency in the text.  

We look forward to receiving your revised manuscript.

Kind regards,

Zacharie Tsala Dimbuene, Ph.D.

Academic Editor

PLOS ONE

Journal Requirements:

3. Please include a caption for figure 1.

Reviewers' comments:

Reviewer's Responses to Questions

**Comments to the Author**

1. Does the manuscript provide a valid rationale for the proposed study, with clearly identified and justified research questions?

Reviewer #1: Yes

Reviewer #2: Yes

2. Is the protocol technically sound and planned in a manner that will lead to a meaningful outcome and allow testing the stated hypotheses?

Reviewer #1: Yes

Reviewer #2: Yes

3. Is the methodology feasible and described in sufficient detail to allow the work to be replicable?

Reviewer #1: Yes

Reviewer #2: Yes

4. Have the authors described where all data underlying the findings will be made available when the study is complete?

Reviewer #1: Yes

Reviewer #2: No

5. Is the manuscript presented in an intelligible fashion and written in standard English?

Reviewer #1: Yes

Reviewer #2: Yes

6. Review Comments to the Author

You may also provide optional suggestions and comments to authors that they might find helpful in planning their study.

Reviewer #1: Some of the statements need to be backed with statistics. For instance, in the abstract it says 'In the sub-Saharan

African region (SSA), the number of adolescents initiated on ART drugs remained consistently low. Although HIV prevalence among adolescents had been reduced, several HIV- positive adolescents died without being initiated on ART drugs.'

How low is low and what is the data for several ALHIV who have died prior to ART initiation.

In abstract methods "A third screener will intervene when disputes arise between the two screeners.' should be reframed to 'A third screener will intervene when there is lack of consensus between the two screeners.'

In manuscript background 'In 2021 alone, approximately 110 000 children and adolescents died from AIDS-related complications in 2021[4].' Is this a global figure or for SSA?

Also 'Although these interventions are prevalent in the sub-Saharan region, HIV infection among adolescents aged 10-19 is still high.' What is the figure?

On page 7 'The study's findings employed healthcare services to revise ART initiation guidelines that will prioritise adolescents' needs and improve healthcare service delivery to accommodate adolescents.' the sentence is not clear and needs to be revised

On page 9 under inclusion criteria- also include types of studies and articles to be included.

Overall the protocol is well written but the authors should pay attention to the grammar in some places and resubmit.

Reviewer #2: A scoping review of factors influencing initiation of antiretroviral treatment among adolescents living with HIV/AIDS in sub-Saharan Africa: A scoping review protocol

This study aims to address a very important topic that would be a valuable addition to the literature. The planned approach (i.e. search in major databases and using PRISMA flowchart) is appropriate. There are generally accepted approaches to scoping reviews and the authors do follow these. it is not very clear however, where the results will be made available

7. PLOS authors have the option to publish the peer review history of their article (what does this mean?). If published, this will include your full peer review and any attached files.

Reviewer #1: No

Reviewer #2: **Yes: **Bareng Aletta S. Nonyane

---

## [Author Response · Author response to Decision Letter 0]

16 Oct 2023

We thank the reviewers and editors for their comments. We have made revisions and attached rebuttal letter, manuscript with track changes and main manuscript.

---

## [Decision Letter · Decision Letter 1]

23 Jan 2024

Mapping factors influencing initiation of antiretroviral treatment among adolescents living with HIV/AIDS in sub-Saharan Africa: A scoping review protocol.

PONE-D-23-19003R1

Dear Ms. Likoti,

We’re pleased to inform you that your manuscript has been judged scientifically suitable for publication and will be formally accepted for publication once it meets all outstanding technical requirements.

Kind regards,

Zacharie Tsala Dimbuene, Ph.D.

Academic Editor

PLOS ONE

Additional Editor Comments (optional):

Reviewers' comments:

Reviewer's Responses to Questions

**Comments to the Author**

1. Does the manuscript provide a valid rationale for the proposed study, with clearly identified and justified research questions?

Reviewer #1: Yes

Reviewer #2: Yes

2. Is the protocol technically sound and planned in a manner that will lead to a meaningful outcome and allow testing the stated hypotheses?

Reviewer #1: Yes

Reviewer #2: Yes

3. Is the methodology feasible and described in sufficient detail to allow the work to be replicable?

Reviewer #1: Yes

Reviewer #2: Yes

4. Have the authors described where all data underlying the findings will be made available when the study is complete?

Reviewer #1: Yes

Reviewer #2: Yes

5. Is the manuscript presented in an intelligible fashion and written in standard English?

Reviewer #1: Yes

Reviewer #2: Yes

6. Review Comments to the Author

You may also provide optional suggestions and comments to authors that they might find helpful in planning their study.

Reviewer #1: All comments have been addressed by the authors and the scoping review protocol in its current form */*is satisfactory.

Reviewer #2: The reviewer comments have been adequately addressed. i agree that the manuscript is improved and should be considered for acceptance.

7. PLOS authors have the option to publish the peer review history of their article (what does this mean?). If published, this will include your full peer review and any attached files.

Reviewer #1: No

Reviewer #2: No

---

## [Editor Report · Acceptance letter]

12 Feb 2024

PONE-D-23-19003R1 

PLOS ONE

Dear Dr. Likoti, 

I'm pleased to inform you that your manuscript has been deemed suitable for publication in PLOS ONE. Congratulations! Your manuscript is now being handed over to our production team.

Kind regards, 

on behalf of

Prof. Zacharie Tsala Dimbuene 

Academic Editor

PLOS ONE